# A Systematic Review to Define the Multi-Faceted Role of Lysine Methyltransferase SETD7 in Cancer

**DOI:** 10.3390/cancers14061414

**Published:** 2022-03-10

**Authors:** Fátima Liliana Monteiro, Cecilia Williams, Luisa A. Helguero

**Affiliations:** 1Department of Medical Sciences, Institute of Biomedicine—iBiMED, University of Aveiro, 3810-193 Aveiro, Portugal; lilianamonteiro@ua.pt; 2SciLifeLab, Department of Protein Science, KTH Royal Institute of Technology, 114 28 Stockholm, Sweden; cecilia.williams@scilifelab.se

**Keywords:** SETD7, cancer, cell proliferation, cell migration, EMT, cell cycle

## Abstract

**Simple Summary:**

The protein methyltransferase SETD7 is essential for epigenetic regulation through methylation of histone H3 and non-histone proteins, including cell cycle, apoptosis and metastasis regulators. This multi-faceted role of SETD7 is cell context-and tissue type-dependent, which makes it difficult to interpret results in the framework of the current literature and to advance research in the field. The aim of this systematic review is to provide an updated description of how SETD7 impacts cancer-related processes considering different cancer types in different cell contexts. In the first part of this systematic review, we characterise the literature and in the second part, we provide a critical assessment of the findings from different cancer types. In the last part of this work, we integrate the findings and summarise the main signalling networks regulated by SETD7 to identify outcomes conserved across studies and propose ways to advance research related to SETD7.

**Abstract:**

Histone–lysine N-methyltransferase SETD7 regulates a variety of cancer-related processes, in a tissue-type and signalling context-dependent manner. To date, there is no consensus regarding SETD7´s biological functions, or potential for cancer diagnostics and therapeutics. In this work, we summarised the literature on SETD7 expression and function in cancer, to identify the contexts where SETD7 expression and targeting can lead to improvements in cancer diagnosis and therapy. The most studied cancers were found to be lung and osteosarcoma followed by colorectal and breast cancers. SETD7 mRNA and/or protein expression in human cancer tissue was evaluated using public databases and/or in-house cohorts, but its prognostic significance remains inconclusive. The most studied cancer-related processes regulated by SETD7 were cell proliferation, apoptosis, epithelial-mesenchymal transition, migration and invasion with special relevance to the pRb/E2F-1 pathway. SETD7 consistently prevented epithelial to mesenchymal transition in different cancer types, and inhibition of its function appears to be associated with improved response to DNA-damaging agents in most of the analysed studies. Stabilising mutations in SETD7 target proteins prevent their methylation or promote other competing post-translational modifications that can override the SETD7 effect. This indicates that a clear discrimination of these mutations and competing signalling pathways must be considered in future functional studies.

## 1. Introduction

Histone–lysine N-methyltransferase SETD7 was first described as a histone H3 lysine 4 (K4) monomethylase. Such monomethylation is often enriched at gene enhancers and associated with active gene transcription [1,2]. Several other non-histone protein targets of SETD7 have since been identified, including numerous transcription factors and epigenetic regulators (reviewed in [3,4]). Methylation by SETD7 modulates protein stability [5], subcellular localisation [6,7], and/or interactions with other proteins [8]. In some cases, methylation by SETD7 competes with other posttranslational modifications to activate or inhibit target protein function [3].

SETD7 expression increases throughout differentiation of human embryonic stem cells [9,10] and cardiomyocyte lineage commitment [11,12], and its histone methyltransferase activity is needed for the chromatin remodelling, which underlies these processes. However, its role in cancer is far from clear as SETD7 expression and activity have been related to both tumour-suppressing and tumour-promoting effects. This may result from the variety of mechanisms and targets regulated by SETD7 and highlights the relevance of cell and tissue context as determinants of SETD7 function.

Initial evidence suggested that methylation of the cellular tumour antigen p53 at K372 by SETD7 promoted p53 stability and function [3]. However, unlike p53^−/−^ mice, which develop tumours after 6 months [13], SETD7^−/−^ mice developed tumours only after 1 year of age [14], and did not show alteration in p53-dependent apoptosis and cell-cycle arrest [15,16]. In fact, other molecules have been shown to act in cooperation with SETD7 to enhance the p53 transcriptional activity and, in absence of SETD7, they can compensate its function [3]. This clearly illustrates the context specificity of the SETD7 function. Moreover, there are several SETD7 inhibitors currently available [17,18,19,20,21,22,23] and some studies have suggested SETD7 inhibition as a potential anti-cancer therapy [20,21,22]. Therefore, there is a need to systematise current findings to identify the contexts that can lead to improvements in cancer diagnosis and therapy.

The aim of this systematic review is to provide an updated description of how SETD7 impacts cancer-related processes in different cell contexts, namely basal growth, upon induction of DNA damage, in hypoxia and oxidative stress. In the first part of this systematic review, we characterise the literature and the risk of bias in the currently available research. In the second part, we provide a critical assessment of findings from different cancer types. In the last part of this work, we integrate findings and summarise the main signalling networks regulated by SETD7 to identify outcomes conserved across studies and propose ways to advance research in this field.

## 2. Materials and Methods

This work adheres to the PRISMA standard for transparent reporting of systematic reviews and meta-analyses [24]. The inclusion criteria comprise all studies focused on SETD7 function and expression in human or mouse cancer tissues or cell lines, in vitro or in vivo, as well as analysis of tissue cohorts or public databases (Table 1). A narrative synthesis method based on Popay et al. [25] was used to describe the data. Gene names were written in italic and according to the NCBI convention [26]. Protein names are not italicised and the recommended protein name comes from UniProt [27], with few exceptions when the alternative name is more widely accepted. All the papers were critically assessed, and any disagreement was resolved by consensus.

### 2.1. Search Strategy

All primary literature reporting SETD7 expression or function in cancer was searched in the PubMed/Medline database using the key terms in Title and Abstract: [(SET7/9 OR SETD7 OR SET7 OR SET9 OR KMT7 OR KIAA1717) AND (cancer OR tumour OR tumor OR carcinoma)]. The search was unlimited, conducted until 5 August 2021 and returned 109 published papers. Existing reviews and reference lists were hand searched for studies missed by the initial query resulting in 2 additional articles.

### 2.2. Eligibility and Data Collection

All retrieved studies were screened for eligibility based on title and abstract analysis, and potentially eligible full-text articles were selected for full-text assessment. Only original reports in English were considered. Reviews, article notes, editorials and comments were excluded. Selected references for which a full-text report was not available after contact with dedicated libraries and with corresponding authors were also excluded. All cell lines used in the remaining studies were accessed on 5 January 2022 using https://web.expasy.org/cellosaurus/ to exclude results using problematic cell lines (misidentified, contaminated by other cell lines, etc.). The inclusion criteria were (1) SETD7 function in non-problematic human or mouse cell lines or mouse cancer models; (2) SETD7 expression in human cancer tissue from in-house cohorts or public datasets properly identified by their accession number; and (3) sufficient data allowing the evaluation of methodological quality (i.e., sample processing, antibodies used, negative and positive controls). From a total of 111 articles (109 from PubMed search plus 2 from other sources), 37 were excluded after title and abstract screening because they were reviews, not related to cancer, or studied PR-Set7 (KMT5A) and one study was excluded because the full-text was not available (Figure 1). Thirty-one studies were excluded after full text analysis because they did not explore the function of SETD7 or did not evaluate its expression in mouse or human cancer cells or tissues (Appendix A). Since 12 of the excluded studies identified SETD7 protein interactions or SETD7 targets (underlined in Appendix A), when appropriate, their findings were integrated into pathway analysis. Data from the remaining 43 articles were extracted using a predefined data collection template (Appendix A).

### 2.3. Quality Assessment

The methodological quality and the risk of bias (i.e., study design, methods and results reporting) were evaluated using the RevMan Risk of Bias assessment tool in *R* and using a list of quality criteria derived from different protocols for systematic reviews [28,29,30]. Risk was categorised as low (green: risk of bias low in all domains), unclear (yellow: risk of bias is unclear in at least one domain, but no domains with high risk), high (red: high risk of bias in at least one domain) or not applicable (white: not applicable to the study in question) (Figure 2 and Appendix A). Any disagreement was resolved by consensus.

## 3. Results

SETD7 was found to either stimulate or inhibit cell proliferation, survival, invasion or metastasis (Appendix A); these processes were grouped as tumour-promoting or tumour-suppressing for analysis purposes (Appendix A). Lung cancer, osteosarcoma, colorectal cancer and breast cancer were the most studied cancer types, where SETD7 function was reported as tumour-promoting and suppressing (Appendix A). Due to the low amount of published data, no contradictory findings were identified in less studied cancers. The outcomes reported in the studies varied, with cell proliferation appearing as the most cancer-related process investigated. To support the mechanistic findings, nearly half of the studies evaluated SETD7 mRNA or protein expression in their in-house patient cohorts or publicly available datasets, with The Cancer Genome Atlas (TCGA) public dataset being the most widely used, followed by in-house cohorts (Appendix A). Overall, the papers included in this work showed low risk of bias, except for methodological bias (bioinformatics) in 40% of the studies using public datasets, which should be considered with caution (Figure 2). All these studies will be described in detail in the following sections, where we describe the main findings in each cancer type. 

### 3.1. SETD7 Expression and Function in Different Cancer Types

#### 3.1.1. Acute Myeloid Leukemia (AML)

Gu et al. [31] screened a panel of six AML cell lines (NB4, HEL, HL-60, THP-1, KG-1a and KG-1) to show that all but KG-1a express *SETD7* mRNA. Knocking-down *SETD7* with shRNAs in KG-1 cells decreased apoptosis after 48 h, but this did not translate into any difference in cell viability measured up to 96 h; while in KG-1a cells, *SETD7* overexpression showed higher apoptosis and reduced viability 72–96 h after transfection. Analysis of *SETD7* mRNA in 35 bone marrow aspirates from AML patients and in 8 from healthy patients showed *SETD7* expression was high in the healthy group and low or weak levels in 18 out of the 35 (51.4%) bone marrow samples from AML patients [31]. On the contrary, in a later study analysing the ONCOMINE database, the same authors found that *SETD7* mRNA expression was upregulated in AML when compared to normal tissues [32]. The differences between these findings might be due to the very small number of healthy controls used and that the ONCOMINE study included data from childhood AML. Zipin-Roitman et al. evaluated SETD7 function using OCI-AML2 cells. They found that the downregulation of N-lysine methyltransferase SMYD2 with shRNAs reduced the number of viable cells, but this was a transient effect partially restored after 5–7 days, which was extended to 12 days if SETD7 activity was also inhibited with 10 µM of (R)-PFI-2. Since *SMYD2* downregulation correlated with SETD7 upregulation (mRNA and protein), a compensatory function of SETD7 in SMYD-deficient cells was proposed. The authors also suggested that *SMYD2* downregulation increases quiescence and hence resistance to DNA damage [33]. However, no experiments showing that the purportedly quiescent cells could originate new colonies were shown. Together, these studies showed that upregulation of *SETD7* reduces cell viability possibly by apoptosis, but this may be a transient effect. In addition, in the context of SMYD2-deficiency, inhibition of SETD7 activity reduces cell viability for longer periods of time. More studies need to be carried out to establish whether *SETD7* expression is different in AML vs. normal samples, and whether SETD7 levels alone or together with those of SMYD2 are associated with the response to genotoxic therapy.

#### 3.1.2. Bladder Cancer

Xie et al. showed that SETD7 protein expression was nearly undetectable in T24 and UM-UC-3 bladder cancer (BlaCa) cell lines as compared to the human normal urothelium cell line SV-HUC-1. The authors studied N6-adenosine-methyltransferase catalytic subunit (hMETTL3), which methylates adenosine residues at the N6 position of some RNAs; this modification is then recognised by the YTH domain-containing family protein 2 (DF2) to regulate mRNA stability. *SETD7* mRNA was reversibly methylated at the N6 position by hMETTL3 and subsequently recognised by DF2 and degraded. Moreover, stable knockdown of *METTL3* reduced the metastatic capacity of the tumours assessed with the tail vein xenograft model. *SETD7* knockdown with siRNAs in both BlaCa lines increased invasion and migration with upregulation of vimentin, cadherin-2/(N-cadherin) and the zinc finger protein SNAI2 (Slug), and downregulation of cadherin-1 (E-cadherin) proteins [34]. This aligns with a previous study in BlaCa NOZ cell line where SETD7 was shown to be a downstream target of the tumour suppressor cadherin-13 (T-cadherin) [35]. On the other hand, overexpression of *SETD7* produced opposite results, and was similar to siRNA knockdown of *METTL3* and *YTHDF2*. Therefore, *SETD7* mRNA is downregulated by hMETTL3/DF2 m^6^A to increase the metastatic potential of BlaCa cells, which supports a tumour-suppressing function of SETD7 [34]. The authors backed their findings with analysis of the TCGA data, where *SETD7* transcript was lower in BlaCa compared with normal samples, with similar levels between stages 2, 3 and 4 [34]. This suggests that loss of SETD7 function could be an early event in the acquisition of the invasive phenotype. 

#### 3.1.3. Breast Cancer

Evidence supporting a tumour suppressor function includes four studies. Montenegro et al. showed that SETD7 inhibits EMT through the upregulation of cadherin-1 and downregulation of vimentin and epidermal growth factor receptor (EGFR) protein levels (Figure 3a). This was found by overexpressing *SETD7* in metastatic, triple-negative MDA-MB-231 cells and downregulating it with siRNAs or inhibiting its activity with (R)-PFI-2 (50 µM; 3 days) in non-metastatic oestrogen receptor α (ERα/*ESR1*)-positive MCF-7 cells. Additionally, in MCF-7 cells, *SETD7* knockdown induced the purported cancer stem-like cell phenotype (CD44^+^/CD24^−/low^) and dedifferentiation of mammospheres associated with loss of cadherin-1. This was in line with more metastasis in MCF-7 xenografts with *SETD7* knockdown [36]. SETD7 methylates DNA (cytosine-5)-methyltransferase 1 (Dnmt1) [37] and E2F-1 [38], inducing their degradation (Figure 3a). Montenegro et al. further showed that stabilisation of E2F-1 induces zinc finger protein SNAI1-mediated transcriptional repression of *SETD7* and *CDH1* genes, and that SETD7 controls *CDH1* expression through enrichment of H3K4me and downregulation of *DNMT1* (Figure 3a). In the cell lines and in a small cohort of BC clinical samples (*n* = 36), *SETD7* mRNA and protein expression negatively correlated with *DNMT1* and *E2F1*. Together, these results indicate that SETD7 has a tumour-suppressing role in BC cells possibly by inhibiting aberrant expression of *E2F1* and *DNMT1*. However, stabilising mutations on *E2F1* and *DNMT1* could potentially override SETD7 function [36]. Kim et al. identified K32 as the hypoxia-inducible factor 1-alpha (HIF-1α) target site for SETD7, with methylation occurring in the nucleus to destabilise HIF-1α and induce its proteasomal degradation. MDA-MB-231 xenografts showed that cells expressing methylation-defective HIF-1α (K32A mutation) resulted in increased tumour formation, tumour weight and volume compared with the cells expressing HIF-1α WT. Hence, suggesting that loss of SETD7-mediated degradation of HIF-1α through methylation may enhance BC growth [39] (Figure 3a). Song et al. used two different clones of the metastatic triple-negative BT-549 and MDA-MB-231 BC cell lines to stably overexpress or downregulate *SETD7*. They found that *SETD7* overexpression reduced cell viability, colony formation and expression of the proliferation marker protein Ki-67, while *SETD7* downregulation exerted the opposite effects and also increased cell migration and invasion. BT-549 BC xenografts overexpressing *SETD7* were smaller than the control cell lines with an empty vector, while the opposite was seen in BC xenografts from BT-549 with *SETD7* knockdown. They also found that by inhibiting the transcriptional activator function of the zinc finger protein GLI1 with cyclopamine, the tumour-promoting phenotype of cells with *SETD7* knockdown was reverted. Since *GLI1* mRNA and protein where negatively correlated with *SETD7* expression, it is possible that tumour-suppressing effects of SETD7 could be mediated by reducing *GLI1* expression (Figure 3a). The same authors also found lower *SETD7* mRNA and protein levels in the BC cell lines BT-549, MDA-MB-231, MCF-7 and MDA-MB-468 compared with non-malignant MCF-10A. When compared with normal breast samples, *SETD7* mRNA was significantly lower in BC (*n* = 79 in each group), especially in the metastatic BC samples [40]. Montenegro et al. compared *SETD7* mRNA levels from 87 samples (77 invasive ductal or lobular carcinomas and 10 benign tumours; all females) and found significantly higher *SETD7* in benign tumours or in tumours that had pathological complete remission, while low *SETD7* mRNA was significantly correlated with reduced overall survival (OS) and disease-free survival (DFS) independently of the tumours’ primary staging and therapy [36]. Lezina et al. also analysed a Genes Expression Omnibus (GEO) public dataset (GSE22226, 221 patients with aggressive biology) to disclose significant correlations between high *SETD7* mRNA with better survival of BC patients [41]. Other authors also analysed public datasets to disclose significant differences in *SETD7* expression in BC samples vs. normal tissue, although no evaluation of *SETD7* correlation with BC prognosis was conducted. Duan et al. found no differences in *SETD7* expression using TCGA data (1097 cancer and 114 normal) [42] and significantly lower *SETD7* expression was detected in BC compared with normal tissues by Gu et al. using the ONCOMINE database [32]. Collectively, these works support the notion that loss of *SETD7* expression accompanies BC progression. Whether this is related to activating EMT as shown in BC cell lines and the relationship to Dnmt1 and E2F-1 protein levels/activity needs to be further explored.

Four studies point towards a tumour-promoting role of SETD7. Zhang et al. used an MCF-7 cell line to show that the erythroid transcription factor (GATA-1) recruits SETD7 to the *VEGFA* promoter, which in turn increases H3K4me1 and Pol II recruitment to the *VEGFA* promoter to increase transcription (Figure 3b). In co-cultures, knocking down SETD7 in MCF-7, MDA-MB-231 and ZR-75-1 cells prevented angiogenesis induced by *GATA1* overexpression in HUVEC cells. SETD7 was needed for GATA-1-induced cell viability in vitro, and for tumour growth and angiogenesis in MDA-MB-231 xenografts. Therefore, SETD7 could mediate GATA-1 tumour-promoting functions in BC [43]. Huang et al. also used MCF-7 and MDA-MB-231 cells to show that stable knockdown of SETD7 vs. empty vector, significantly reduced cell viability already after 2 days and up to 4 days, which could be due to increased apoptosis (Annexin V/PI). *SETD7* knockdown increased ROS levels by 3.5-fold and reduced the GSH/GSSG ratio, which was related to the upregulation of the Kelch-like ECH-associated protein 1 (KEAP1) and downregulation of nuclear factor erythroid 2-related factor 2 (Nrf-2) protein, as well as Nrf-2 target gene *ME1*, *TXNRD1*, *GCLC* and *GCLM* expression (Figure 3b). Therefore, SETD7 may be needed to maintain redox homeostasis and its loss induces apoptosis by increasing oxidative stress. However, this study did not control for off-target shRNA effects. Analysing data from the TCGA database (*n* = 1086), *SETD7* expression was strongly and negatively correlated with *KEAP1*; but positively correlated with *ME1*, *TXNRD1*, *GCLC* and *GCLM*. Additionally, lower *SETD7* expression predicted better OS in patients with low expression of the Nrf-2 target genes, which suggests that SETD7 protects BC cells from oxidative stress through downregulation of *KEAP1* [44]. Si et al., using the same cell lines, found that knockdown of *SETD7* with shRNA reduced growth rate, colony formation and tumour volume. This correlated with the decreased expression of *RUNX2*, which they showed to be a direct SETD7 target gene by using chromatin immunoprecipitation sequencing (ChIP-seq). Overexpression of *RUNX2* in *SETD7* knockdown cells partially restored the phenotype. They also showed that E3 ubiquitin-protein ligase TRIM21 interacts with the middle terminal domain of SETD7 inducing its proteasomal degradation (Figure 3b). Therefore, failure to inactivate SETD7 through the proteasome may result in enhanced expression of its target gene *RUNX2*, and consequently proliferation, migration and invasion of BC cells [45] (Figure 3b). Comparing *SETD7* mRNA and protein levels in a panel of BC cell lines (T-47D, MCF-7, UACC-812, MDA-MB-231 and MDA-MB-468) with MCF-10A non-malignant mammary epithelial cells, Si et al. found higher SETD7 levels in the BC cells [45]. Notably, this is opposite to Song et al., as described above [40]. It has been shown that SETD7-mediated methylation of ERα at K302 stabilises the receptor and enhances the expression of its target genes [46]. Takemoto et al. used the SETD7 inhibitor cyproheptadine (43 μM), and transient SETD7 knockdown, to show that SETD7 downregulated ERα protein stability by decreasing its half-life, which was in agreement with reduced *ESR1* transcriptional activation and impaired oestrogen-dependent cell viability in MCF-7 cells (Figure 3b). In vivo, cyproheptadine for 8 days (either single or continuous intraperitoneal administrations at 20 mg/kg) significantly inhibited the tumour growth of MCF-7 xenografts [21]. However, no studies were carried out in vivo to effectively show that the effect of cyproheptadine was through reducing ERα stability.

**Figure 3 cancers-14-01414-f003:**
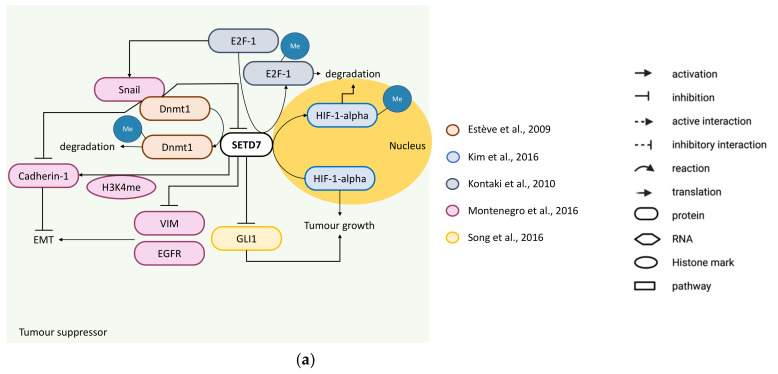
SETD7 targets and their regulation of cancer-related processes in breast cancer. (**a**) Tumour suppressor—SETD7 can inhibit EMT through methylation of Dnmt1 [37] and E2F-1 [38] inducing their proteasomal degradation to decrease snail expression and consequently release inhibition of cadherin-1 and SETD7 expression, and impair the snail/Dnmt1 feedback loop. SETD7 can also upregulate cadherin-1 directly through H3K4me enrichment at its promoter and inhibit vimentin and EGFR expression preventing EMT [36]. SETD7 can inhibit tumour growth through methylation of HIF-1α to promote its degradation [39] or by inhibiting GLI1 expression [40]. (**b**) Tumour promoter—SETD7 can induce cell proliferation through RUNX2 upregulation [45] or ERα methylation [21]. SETD7 promotes GATA1-induced VEGFA expression to stimulate migration, invasion and angiogenesis [43]. SETD7 can also protect breast cancer cells from oxidative stress by reducing reactive oxygen species (ROS) via inhibition of KEAP1 and stimulation of GSTT2 and Nrf-2 expression through H3K4me enrichment in their promoters, and supporting the expression of Nrf-2-target genes (TXRND1, ME1, GCLC, GLCM and NQO1) to prevent ROS [44].

Several studies support a tumour-promoting role of SETD7 through the analysis of in-house cohorts and databases. Zhang et al. carried out SETD7 immunohistochemistry (IHC) in 80 human BC samples and found higher SETD7 in BC tissues vs. non-tumorous tissue, although these were not paired samples. SETD7 protein expression was positively associated with tumour size and grade and nodal status. In line with this, higher SETD7 levels were correlated with shorter DFS and OS and were demonstrated to be a significant marker of poor prognosis for both DFS and OS. SETD7 and ERα were inversely correlated [43]; however, the analysis pooled all BC subtypes. Si et al. analysed data from the Human Protein Atlas database and showed that SETD7 exhibited stronger positive expression in BC tissues when compared with normal breast, while mRNA expression data from the GEO database (GSE9893: 132 primary tumours from tamoxifen-treated patients; GSE12276: 204 primary tumours from patients with known site of relapse) revealed that *SETD7* expression negatively impacts patient OS rates and local relapses, which is in agreement with their findings using the online Kaplan–Meier plotter [45]. Huang et al. analysed TCGA data (*n* = 1086) and reported that high *SETD7* expression was significantly associated with higher *ESR1* and *PGR* mRNA expression and lower *ERBB2*. Higher *SETD7* expression was significantly associated with poorer OS (being the median survival time of patients with low *SETD7* expression (18.1 years) twice than that of *SETD7* high-expressed patients (9.5 years)). However, multivariate analysis did not support SETD7 as an independent prognostic factor for BC [44]. No information on whether the analysis was carried out by BC subtypes, or all types is available in any of the studies analysing public datasets, although it appears by the number of cases, that all subtypes were included.

In summary, differences in *SETD7* mRNA or protein levels in BC cell lines vs. non-malignant cell lines may relate to methodological bias (i.e., antibody specificity, confluency, time in culture before sample processing, effect of transfection reagents on basal cellular stress as shown previously [47] and also observed by us (unpublished)). The correlation studies used a variety of cohorts and different antibodies when analysing SETD7 protein expression by IHC [36,45]. The high risk of bias regarding characterisation of public databases [32,41,42,45] or the comparison of BC tissue with [36,40] or BC adjacent non-tumoral tissues as controls [43], or whether BC data were analysed as a whole or in selected subtypes does not allow a conclusion and highlight the need of future studies. Importantly, the functional studies showed that the tumour-suppression role of SETD7 is related to its methylation of oncogenic target proteins (Dnmt1, E2F-1 and HIF-1α) leading to their degradation. On the contrary, SETD7 may be important to prevent oxidative stress either by reducing *KEAP1* and enhancing the expression of *GATA1* and *VEGFA,* which together with induction of *RUNX2*, could promote metastasis. Finally, SETD7 control of ERα stability may be of relevance to endocrine resistance and deserves future studies.

#### 3.1.4. Cervical Cancer

HeLa cells were used to study the SETD7 role in cervical cancer cell line [48,49]. Shen et al. showed that catenin beta-1 (Beta-catenin) is methylated by SETD7 at K180 and that this destabilises Beta-catenin and leads to its proteasomal degradation. Knockdown of *SETD7* by shRNAs or Beta-catenin unmethylatable mutation (K180R) significantly enhanced the expression of Wnt/Beta-catenin target genes *MYC* and *CCDND1* and promoted cell proliferation [48]. Wu et al. showed that the transcription factor YY2 is monomethylated by SETD7 at K247. By ChIP-seq analysis they found K247me promoted YY2 binding to DNA and transcription of *ABL1*, *TP53*, *RAD1*, *CCNT2* and *CCNA2* genes. Two YY2 somatic mutations (K244Q and S246F) near K247 reduced its methylation by SETD7, while the unmethylatable K247R mutant was not as efficient in attenuating cell proliferation and tumour growth. Therefore, SETD7-mediated methylation of YY2 at K247 positively regulates YY2-target gene’s transcription, which mediates the inhibitory function of YY2 in cell proliferation and tumour growth [49], and points to mutations affecting K247 methylation that could be pathologically relevant.

In summary, the two studies described that tumour-suppression by SETD7 results from its methylation of oncogenic proteins leading to their degradation. Hence, this effect could potentially be overridden by stabilising mutations.

#### 3.1.5. Colorectal Cancer

Evidence supporting a role for SETD7 as tumour suppressor was presented in six studies (Figure 4). Xie et al. studied the regulation of E2F-1 by SETD7 in cells undergoing cell death induced by DNA damage (20–50 mM cisplatin for 24 h) and found that SETD7 methylates E2F-1 at K185, stabilising it to upregulate the protein levels of its pro-apoptotic target genes, *BIM1* and *TP73* (Figure 4). Overexpressing SETD7 in the p53-negative HCT116 cell line promoted DNA damage-induced cell death and reduced cell number; while *SETD7* knockdown with siRNAs had the opposite effect, and a SETD7 mutant (H297A), which lacks methyltransferase activity, also increased the cell number upon DNA damage. Therefore, when DNA damage is induced, SETD7 monomethylates E2F-1 at K185 to promote its pro-apoptotic signalling independently of p53 [50]. Liu et al. showed that SETD7 protein levels were increased by resveratrol and were associated to decreased cell viability in HCT116, Co-115 and SW48 cell lines [51]. SETD7 monomethylates p53 at K372 to increase its stability and pro-apoptotic activity [52] and resveratrol increased p53 K372me levels (Figure 4). Additionally, SETD7 stable knockdown with shRNA for 48 h attenuated the resveratrol-induced p53 protein overexpression. SETD7 knockdown also decreased cleaved PARP-1 and caspase-3. Therefore, induction of SETD7 by resveratrol promoted CRC cell apoptosis possibly in a p53-dependent manner [51]. Hong et al. showed that SETD7 methylates the serine/threonine-protein kinase RIO1 at K411 to induce its ubiquitination and degradation through the F-box only protein 6 (FBXO6) (Figure 4). RIO1 promotes the proliferation, invasion and metastasis of CRC cells in vitro and in vivo and its phosphorylation at T410 by Casein kinase stabilises RIO1 because it blocks SETD7-mediated K411me. HCT116 cells stably expressing the RIO1 unmethylatable K411R mutation or K411R plus phosphomimetic mutation (K411R/T410E), had higher proliferation, migration and invasion compared to RIO1-WT cells, whereas cells stably expressing a non-phosphorylatable mutation (T410A) of RIO1 inhibited these processes. This translated into larger and more metastatic tumours in the K411R cells or retarded growth in the T410A cells as compared to RIOK-WT cells. Therefore, methylation of RIO1 by SETD7 leads to its degradation to suppress CRC cell growth and metastasis but can be counterbalanced by RIO1 phosphorylation (Figure 4). Consistently with their in vitro results, when analysing a dataset of 104 CRC by IHC, an inverse expression pattern between RIO1 and SETD7 or FBXO6 was observed. Additionally, low levels of SETD7 and FBXO6 in CRC samples correlated with poor OS [53]. Wang et al. showed that overexpression of *SETD7* in the HCT116 cell line positively regulated the expression of goblet cell differentiation marker *ATOH1* (Figure 4), whereas *SETD7* knockdown by siRNA for 48 h increased cancer stem cell markers *NANOG* and *CD24*. They also showed that *SETD7* is a target of miR-372/373 (Figure 4), which is a cluster of stem cell-specific microRNAs that are often upregulated in CRC. These results emphasise SETD7 role in promoting CRC differentiation in detriment of the cancer stem cell phenotype [54]. Zhang et al. showed that *SETD7* overexpression in HCT116 and SW480 cell lines reduced cell number and migration together with increased acetylated α-tubulin and the opposite was seen when *SETD7* was knocked down by siRNA. SETD7 directly interacted with histone deacetylase 6 (also known as HDAC6) suppressing its deacetylation activity which explained the increase in acetylated α-tubulin (Figure 4). These results suggest that SETD7 may play a tumour-suppressing role in CRC by inhibiting HDAC6 deacetylation activity [55]. Vasileva et al. found that SETD7 interacts with the cytoplasmic KH domain-containing, RNA binding, signal transduction-associated protein 1 (also known as Sam68) and methylates its K208, which might increase its cytoplasmic localisation (Figure 4). Sam68 nuclear activity has been linked to the development of several types of carcinomas, including CRC. CRISPR/Cas9 knock-out of *SETD7* in HCT116 cells decreased the cytoplasmic Sam68 protein and induced cell accumulation in early S phase and apoptosis. The authors suggested that SETD7 in CRC cells affects Sam68 cell localisation, neutralising its nuclear function [6], although this may also be due to total protein reduction. The authors supported their conclusions by analysis of *SETD7* expression in the GSE39582 dataset (443 CRC samples) using the Syntarget tool (bioprofiling.de) to show that high co-expression of *SETD7* and *KHDRBS1* (Sam68) significantly correlated with better overall survival of CRC patients, while patients with low *SETD7* and high *KHDRBS1* expression showed poor OS. However, the expression levels of *KHDRBS1* or *SETD7* alone did not display any statistically significant correlation with survival rates of cancer patients [6]. Liu et al. showed that low expression of *SETD7* is associated with lower survival rates when analysing the GSE17537 dataset (*n* = 55 CRC cases) [51]. Another study assessed SETD7 by IHC in 54 CRC tissues and found it was significantly lower compared with adjacent tissue. Additionally, low levels of SETD7 in CRC correlated with poor OS [55].

Two studies presented evidence of SETD7 as a tumour promoter. Francis et al. developed an SETD7 methyltransferase screening assay and found that the compound 610930-N (2,8-bis(2,4-dihydroxy-6-methylphen ylamino)-1,9-dimethyl dibenzo[b,d]furan-3,7-dione) inhibits SETD7 activity. Incubation of HCT116 cells with 610930-N (52-104 µm for 24-48h) inhibited proliferation and reduced H3K4me levels, which serves as marker that the effect was likely through inhibition of SETD7 activity [18], but does not rule out other off-target effects. Duan et al. showed that SETD7 knockdown with siRNA in HCT116 and RKO cell lines reduced cell viability through G1/S cell cycle arrest and apoptosis [42]. An exploratory proteomic profiling analysis on serum from paired pre-and post-operative CRC patients, colorectal polyps patients and healthy controls (*n* = 24 per group), showed that SETD7 better distinguished CRC patients from healthy controls and patients with colorectal polyps, being in higher abundance in serum from CRC patients and significantly diminishing after tumour removal. This was validated by ELISA in 115 paired pre- and post-operative CRC patients, 38 patients with colorectal polyps, and 38 healthy controls, and by IHC in 176 CRC, 20 colorectal polyps and 20 para-cancerous tissues from colon CRC patients. These results correlated with the downregulation of *SETD7* expression in normal colorectal mucosa compared to CRC tissue from the TCGA (380 cancer and 51 normal) database [42], which is in accordance with Gu et al. analysis also using the TCGA dataset [32]. However, there is high risk of bias related to detailed methodological analysis of these studies, respectively.

In summary, all except one functional study showed that downregulation of *SETD7* activates cellular signalling towards proliferation, which might be related to the methylation of SETD7 target proteins leading to their degradation (RIO1) or stabilisation (E2F-1, p53). All studies used similar cell growth conditions and methodological approaches, while off-target effects of the inhibitor cannot be ruled out in Francis et al. [18]. Differences in the expression studies could be related to different antibody and/or comparing CRC with healthy controls and benign disease instead of adjacent non-tumoural tissues in Duan et al. [42]. The unclear [6,42,51] or high [32] risk of bias regarding characterisation of public databases does not allow a conclusion and highlights the need of future studies.

#### 3.1.6. Epithelial Ovarian Cancer

Zhou et al. identified SETD7 as a target of miR-153. In OVCAR-3 cell line, *miR-153* overexpression inhibited proliferation, invasion and migration, and the miR-153 inhibitor produced opposite results. Both phenotypes were rescued by *SETD7* overexpression or knockdown, respectively. *miR-153* mRNA was lower in OVCAR-3 and SK-OV-3 cell lines compared to normal ovarian cell lines (NOE095 and hOSEpiC), and *SETD7* expression was higher in cancer cells. Concordantly, analysis of 60 cases and their adjacent normal tissue showed an inverse correlation between miR-153 and *SETD7* mRNA expression, with *miR-153* downregulation in the cancer tissue. Additionally, Kaplan–Meier curves plotted from clinical data showed that higher expression of SETD7 indicated a worse 5-year survival [56].

#### 3.1.7. Gastric Cancer

Hong et al. showed that RIO1 methylation by SETD7 promotes its ubiquitination and degradation in the MKN45 cell line. This suggests that SETD7 may inhibit RIO1-mediated proliferation, invasion and metastasis [53]; no studies were carried out to validate this. Akiyama et al. showed that knockdown of *SETD7* with siRNAs increased cell proliferation in MKN45, AGS and KATO-III cell lines, and migration and invasion in MKN74, MKN45 and AGS cells. Using microarrays, they showed that *SETD7* knockdown downregulated *SREK1IP1*, *PPARGC1B* and *CCDC28B* and upregulated *MMP1*, *MMP7* and *MMP9*. The regulation of *SREK1IP1*, *PPARGC1B* and *CCDC28B* genes seemed to be direct through enrichment of H3K4me1 at their promoters. These results suggest that loss of SETD7 may contribute to GC cells proliferation and invasion and is supported by lower levels of *SETD7* mRNA and/or protein in a panel of 12 GC cell lines (AGS, GCIY, HSC-43, HSC-44PE, HSC-57, HSC-60, NUGC-4, KATO-III, MKN7, MKN45, MKN74 and TGBC11TKB) compared with three non-cancerous gastric mucosa samples, by RT-PCR and WB. Additionally, SETD7 protein expression in 376 primary GC tissues by IHC showed loss or weak expression in GC compared with matched non-cancerous tissue, which was associated with age, gender, Lauren’s classification, Ming’s classification, perineural invasion, pT stage, lymph node metastasis and pTNM stage. Interestingly, a lower SETD7 total score in advanced GCs was more significant than that in early GCs. Overall, reduced SETD7 expression was significantly correlated with clinical aggressiveness and worse prognosis, including lower OS and DFS. An additional cohort of 25 other primary GCs tissues was used to compare SETD7 expression at mRNA and protein levels and they both correlated [57]. Other studies have also analysed *SETD7* mRNA in public databases and found downregulation of *SETD7* transcript in GC (TCGA: 415 cancer and 35 normal) associated with shorter OS [42].

In summary, SETD7 prevents cell proliferation and migration; its expression is reduced in GC compared to non-cancerous cells and tissues, and its mRNA and protein negatively correlate with OS. 

#### 3.1.8. Glioma

Li et al. found that *SETD7* knockdown by siRNA in U251 cells decreased the expression of pro-apoptotic proteins (BAX and cleaved caspase-3) whereas the expression of the anti-apoptotic *BAD* was increased, which significantly increased cell proliferation, invasion and migration. They further proposed that these effects were mediated by a concomitant loss of H3K4me3 on the promoter of the long non-coding RNA DRAIC to reduce its expression and thus prevent DRAIC suppression of miR-18a-3p expression. The authors also analysed *SETD7* levels in 20 glioma and neighbouring normal tissues, with lower mRNA in glioma, while Kaplan–Meier curves plotted from clinical data showed that high expression of *SETD7* was associated with better overall survival, suggesting SETD7 has a tumour-suppressor role in glioma [58].

#### 3.1.9. Hepatocellular Carcinoma

Data supporting a tumour-suppressor role of SETD7 comes mainly from Wang et al. who showed that in the HCCLM3 cell line, the long non-coding RNA SNHG6 interacts with *SETD7* mRNA, via the hnRNP L binding protein to destabilise it. Thus, knockdown of SNHG6 increased SETD7 protein levels by suppressing SETD7 mRNA degradation, while SETD7 overexpression in HCCLM3 cells decreased cell migration and invasion, decreased the protein levels of SNAI2 and vimentin and increased cadherin-1; *SETD7* knockdown by siRNA in Huh7 cells produced opposite results [59]. The same authors showed a downregulation of *SETD7* mRNA in 100 HCC tissues compared to adjacent tissues and even lower *SETD7* mRNA levels in HCC complicated with portal vein tumour thrombus. Duan et al. also analysed *SETD7* transcript in HCC compared with normal tissues using public datasets and reported lower *SETD7* in tumours (TCGA; 371 HCC and 50 normal) [42]. SETD7 may act as a tumour suppressor in HCC [59].

Evidence of SETD7 as a tumour promoter comes from two studies. Gu et al. found that SETD7 regulation of E2F-1 through protein–protein interaction in Huh-7 cells occurs in the cytoplasm, but not in the nucleus. Knockdown of *SETD7* with shRNAs decreased E2F-1 protein (but not its transcript) and E2F-1 downstream targets (*CCNA2*, *CCNE1* and *CDK2*), suggesting that SETD7 increases E2F-1 stability. Functionally, downregulation of *SETD7* either by shRNA or the pan-protein methylation inhibitor 5’-deoxy-5’-methylthioadenosine, significantly reduced cell proliferation, migration and invasion, which appeared to be through reduced E2F-1 as similar results were obtained when silencing it. The same authors analysed a cohort of 68 HCC tissues compared to the paired adjacent healthy tissues by IHC, and found SETD7 significantly correlated with pathological stage and tumour size [32]. Chen et al. also found high *SETD7* mRNA and protein expression significantly correlated with metastasis, recurrence, poor tumour differentiation, large tumour size and shorter OS. They showed this using two approaches: bulk analysis of 20 pairs of HCC and adjacent non-tumour tissues by PCR and WB, and by IHC of 225 pairs of paraffin-embedded tissues and adjacent non-tumour tissues [60]. Collectively, these results suggest that SETD7 may have a tumour-promoting role in HCC by stimulating the cell cycle through E2F-1 stabilisation and possibly also by enhancing migration and metastasis [32].

In summary, SETD7 may decrease cell migration and invasion via downregulation of SNAI2 and vimentin and upregulation of cadherin-1 and promote E2F-1-mediated cell proliferation. The differences from functional studies could not be explained by any methodological bias, but as explained above, may result from different experimental conditions such as cell confluency or basal cellular stress caused by the transfection protocols used. In the expression studies, Gu et al. [32] and Chen et al. [60] analysed their in-house cohort using IHC while Wang et al. [59] and Duan et al. [42] analysed mRNA with the later showing high bias in methodological description of the public database and analysis. 

#### 3.1.10. Lung Cancer

Studies supporting a tumour-suppressor role of SETD7 include four papers. Kim et al. identified K32 as the HIF-1α target site for SETD7, with methylation occurring in the nucleus to destabilise HIF-1α and induce its proteasomal degradation (Figure 5a). Knock-in mice bearing a methylation-defective *Hif1a*^KA/KA^ allele exhibited increased lung tumour growth, vascularisation and angiogenesis. They showed that two somatic mutations of HIF-1α (S28Y and R30Q) occurring near the K32 in several human cancers, including LCa, impair the methylation-dependent degradation of HIF-1α induced by SETD7 [39]. A second methylation target site of SETD7 on HIF-1α protein (K391) also destabilises HIF-1α but can be reversed by the lysine-specific histone demethylase 1A (also known as LSD1) to induce *VEGFA* transcription and tumour angiogenesis [61]. Together, these results demonstrate that SETD7 may act as tumour suppressor by negatively modulating HIF-1α-, but this negative regulation may be impaired if LSD1 is also highly expressed/active [39] (Figure 5a). Wang et al. studied histone-lysine N-methyltransferase SUV39H1, which catalyses the H3K9m3 to compact and stabilise heterochromatin. In response to DNA damage by irradiation (UV-C 60 J/m^2^) or chemotherapy (10 μM cisplatin for 12 h or 0.5–1 μM adriamycin for 12–72 h), SETD7 inhibited SUV39H1 methyltransferase activity through K105 and K123 methylation, resulting in heterochromatin relaxation and genome instability in H1299 cells (Figure 5a). Additionally, it has been shown that SETD7 methylation of SUV39H1 resulted in the inhibition of mouse embryonic fibroblast proliferation [62]. Daks et al. showed that, in normal conditions, knockout of *SETD7* by CRISP/Cas9 in A-549 cells (*KRAS* mutated and high expression of *SETD7*) or knockdown by shRNAs in H1299 cells (wild-type *KRAS, TP53* null and low expression of *SETD7*) resulted in an increased cell number and an accumulation of cells in S-phase and a decrease in cells in G1-phase with a concomitant increase in *CCNA1* and *CCND1* expression (Figure 5a). Cadherin-1 protein levels were reduced by the ablation of SETD7 to increase cell migration [63]. Finally, Cao et al. knocked-down or overexpressed *SETD7* in A-549, H1299 and H661 cells. Cell viability was not affected by *SETD7* downregulation with siRNAs, but this promoted cell migration and invasion, which was related to upregulation of the metastasis-related transcripts (*MMP2*, *TWIST1* and *VEGFA*), increase in cadherin-2 and vimentin and decrease in cadherin-1 proteins and migration related to activation of the JAK2/STAT3 pathway. On the other hand, *SETD7* overexpression reduced the JAK2/STAT3 pathway and inhibited migration and invasion, which was in line with downregulation of metastasis-related mRNA transcripts (Figure 5a), but had no effect on cadherin-2, vimentin or cadherin-1 protein levels. LCa cell lines (H661, A-549 and H1299) exhibited lower SETD7 levels compared with non-malignant human bronchial epithelial cells (BEAS-2B) [64] and analysis of *SETD7* mRNA levels in 10 pairs of LCa tissue with matched noncancer tissues showed that *SETD7* was significantly lower in LCa tissues. This result was corroborated by SETD7 WB in three pairs of tissues [64]. In accordance, Duan et al. analysed *SETD7* expression in the TCGA dataset (1017 cancer and 110 normal samples) and reported a significant downregulation of *SETD7* in LCa [42].

The results pointing towards a tumour-promoting role of SETD7 come from seven studies. Kontaki et al. showed that overexpression of *SETD7* in the p53-null H1299 cell line decreased the percentage of cell death upon DNA damage induced by 50 mM etoposide or 0.5 mM doxorubicin for 24 h. In contrast, *SETD7* knockdown or overexpression of a mutant SETD7 lacking methyltransferase activity, induced cell death and upregulation of E2F-1. The effect seems to be mediated by E2F-1, since upon DNA damage, E2F-1 directly stimulated *TP73* transcription and *E2F1* knockdown decreased cell death by doxorubicin and abrogated the inhibition of apoptosis by SETD7. Moreover, upon induction of DNA damage, methylation of E2F-1 in K185 by SETD7 prevented acetylation and phosphorylation of E2F-1, which are needed for its stability and pro-apoptotic function [38] (Figure 5b). This is supported by findings from Gu et al. who also showed that *SETD7* knockdown by shRNAs increased apoptosis of the p53-null H1299 and A-549 cell lines via upregulation of E2F-1, whereas SETD7 overexpression significantly downregulated E2F-1 and decreased apoptosis independently of ectopic expression of p53 alone or with 0.5 µmol/L adriamycin for 12 h [31]. Lezina et al. published two studies on LCa cells undergoing DNA damage by 0.5 µM doxorubicin up to 48 h or 5 gray of gamma irradiation up to 16 h [41,65]. In the first study, they induced G1/S-phase arrest through stable knockdown of *SETD7* by shRNAs in H1299 cells, either with or without DNA-damaging agents to prevent *CCNE1* expression via E2F-1, independently of *TP53* status, as shown in a panel of six cell lines (H23, H520, H522, H1650, H1299 and H460) [65]. Moreover, analysis of two different LCa GEO datasets (GSE31210 and GSE36471) showed that a positive correlation between *SETD7* and *TP73* positively corelates with poor survival compared with no or negative correlation between these two genes, which showed better patient survival [65]. These results indicate that even though SETD7 downregulates E2F-1 activity preventing *TP73*-mediated apoptosis, SETD7 co-activates E2F-1- dependent transcription of *CCNE1* to induce proliferation (Figure 5b). Interestingly, in the second study, Lezina et al. showed that high SETD7 (H522 cell line) and low SETD7 (H1650 cell line) differed in their response to DNA damage by doxorubicin, with high SETD7 reducing the sensitivity, and low SETD7 associated to extreme sensitivity [41]. The Kontaki et al. [38] and Lezina et al. studies [41,65] focussed on two different processes regulated by E2F-1, and while they appear contradictory as to how SETD7 may impact E2F-1 activity, they both support SETD7 inhibition to improve sensitivity to DNA-damaging agents, which is in agreement with a more recent study by Daks et al. [63] who showed that under genotoxic stress conditions induced by doxorubicin or etoposide, *SETD7* knockout or knockdown, or methyltransferase activity inhibition by 2 mM (R)-PFI-2 resulted in increased sensitivity to chemotherapy with higher cell death and lower cell viability. (R)-PFI-2 also increased the sensitivity of primary-patient derived non-small LCa cells to doxorubicin, thus supporting the idea that SETD7-selective inhibitors could be used to enhance anti-cancer therapy [63]. Fu et al. found that SETD7 positively regulates the activation of SHH signalling by methylating the full-length transcription activator GLI3 at K436 and K595 to improve GLI3 stability (K436) and DNA binding to the promoter regions (K595) of its downstream target GLI1. This was not observed with the truncated GLI3^R^ protein. A-549 cells expressing unmethylatable GLI3 mutants had their proliferation, migration/invasion and tumourigenesis capacities compromised. Since both *SETD7* and *GLI1* mRNA levels were upregulated in A-549 cell line when compared to the non-malignant bronchial epithelial lung cell line BEAS-2B, it is likely that SETD7 activates the SHH signalling in LCa promoting tumour growth and metastasis in vitro and in vivo via GLI3 methylation [66] (Figure 5b). Meng et al. found that the long non-coding RNA *LINC01194* acts as a tumour promoter in lung adenocarcinomas by positively modulating *SETD7* expression (Figure 5b) through competitive binding to miR-641. *SETD7* knockdown by shRNAs in A-549 and HCC827 cell lines led to a decreased cell proliferation, migration and invasion and an increase in cell apoptosis rate. These results correlated with the upregulation of *SETD7* in LCa cell lines (A-549, PC-9, HCC827, NCI-H1975 and H1299) compared with the normal human bronchial epithelioid cell line, 16HBE. The same overexpression pattern of *SETD7* was observed in LCa vs. normal tissue (64 pairs of lung adenocarcinoma and adjacent non-tumour tissues) [67], which is in agreement with data analysed by Gu et al. [32]. According to Daks et al., high *SETD7* mRNA expression was associated with shorter survival probability (GSE11969—149 patients with non-small cell LCa, including 90 patients with adenocarcinomas) [63].

In summary, under stress conditions, SETD7 may act as a tumour promoter reducing the response to therapy of LCa cells, since all studies [31,38,41,63,65] except one [62] found that downregulation of *SETD7* or the inhibition of its activity increases sensitivity to chemotherapy. This might be through the regulation of E2F-1 and be independent of p53 status. On the other hand, under non-stress conditions, SETD7 seems to act as tumour suppressor since its downregulation increases the activation of JAK/STAT3 signalling to stimulate cell proliferation, migration [63] and invasion [64] pathways, as well as enhances HIF-1α stability to promote angiogenesis and tumour growth [39]. Additionally, all correlation studies [32,63,65,67], except two [42,64] found *SETD7* to be higher in LCa tissues when compared to adjacent non-tumorous tissues and one further correlating high expression of *SETD7* with worse prognosis. The differences in the correlation studies probably relate to different methodologies and cohorts [32,42,63,65] since the methodology used for public database analysis was poorly described and the cohort used by Cao et al. [64] was very small (only 10 samples). Thus, future studies are needed to confirm that *SETD7* expression is higher in LCa compared with non-malignant tissue.

**Figure 5 cancers-14-01414-f005:**
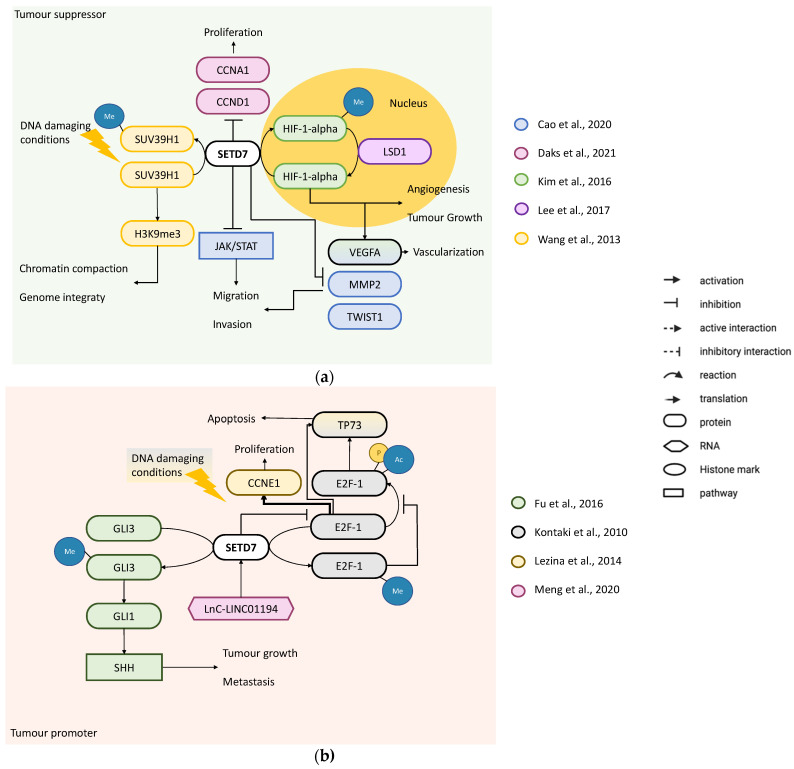
SETD7 targets and their regulation of cancer-related processes in lung cancer. (**a**) Tumour suppressor—SETD7 can inhibit proliferation through downregulation of CCNA1 and CCND1 [63]. SETD7 can also inhibit angiogenesis and tumour growth through methylation of HIF-1α [39], which can be demethylated by LSD1 to induce VEGFA [61]. SETD7 inhibits MMP2, TWIST1 and VEGFA and/or JAK/STA pathway [64] to prevent migration. Under DNA damaging conditions, SETD7 methylates SUV39H1 inhibiting its methyltransferase activity, which results in genome instability and consequent apoptosis of LCa cells [62]. (**b**) Tumour promoter—SETD7 methylates GLI3 to improve its stability and DNA binding to the promoter regions of its downstream target GLI1 which positively regulates the activation of SHH signalling in LCa promoting tumour growth and metastasis [66]. The long non-coding RNA *LINC01194* acts as a tumour promoter in lung adenocarcinomas by positively modulating *SETD7* expression [67]. Under DNA damaging conditions, SETD7 directly methylates E2F-1 preventing E2F-1 stabilisation by acetylation to promote apoptosis mediated by TP73 [38]. SETD7 downregulates E2F-1 activity, thus preventing TP73-mediated apoptosis; moreover, SETD7 co-activates E2F-1-dependent transcription of *CCNE1* [65].

#### 3.1.11. Osteosarcoma

The tumour-suppressor role of SETD7 was reported by four studies. Ivanov et al. used knockdown of *SETD7* with shRNA in U2OS cells treated with 0.5–2 mM adriamycin for 1.5–24 h to induce DNA damage, and showed that p53 was destabilised together with the downregulation of *CDKN1A* expression, which impaired p53-dependent cell cycle arrest in G2 phase [52] (Figure 6a). Since methylation of p53 by SETD7 at K372 stabilises it to enhance transcription of *CDKN1A* (also known as p21) and *MDM2* [68,69], this may be critical for p53 activation in response to DNA damage in osteosarcoma cells [52]. This is aligned with results in mouse embryonic fibroblasts from SETD7^−/−^ mice that lack p53 methylation and failed to arrest following DNA damage [14]. Xie et al. overexpressed *SETD7* in U2OS cells to enhance cell death by DNA damage (20–50 mM cisplatin for 24 h), while *SETD7* knockdown with siRNA resulted in the opposite. SETD7 methylates E2F-1 at K185 stabilising it to enhance the transcription of pro-apoptotic target genes, *BIM1* and *TP73*, suggesting that SETD7 activity inhibits cell growth and enhances the E2F-1 pro-apoptotic effect in osteosarcoma cells [50] (Figure 6a). pRb is a negative regulator of the cell cycle by inhibiting E2F transcription factors [70]. Munro et al. found that pRb was stabilised by SETD7 through methylation at K873, which results in repression of E2F-1 transcription by *CDC6*, *DHFR*, *CDC25A* and *CDK1* and consequent G1/S arrest (Figure 6a). SaOS-2 cells expressing a mutant form of pRb (K873A) had lower accumulation in G1 and lower number of senescent cells as compared to expression of wild type pRb [71]. Another study from the same group further investigated the pRb methylation role in cell cycle arrest of U2OS cells treated with chemotherapeutic agents (10 mM etoposide for 16 h). They found that pRb can be methylated by SETD7 also at K810, a known conserved basic residue (SPXK) required for cyclin-dependent kinase recognition and phosphorylation, thereby retaining pRb in the hypophosphorylated growth-suppressing state. Hence, *SETD7* knockdown in U2OS cells resulted in increased levels of phosphorylated pRb and the E2F-1 targets, dihydrofolate reductase, cyclin-dependent kinase 1 (CDK1) and cell division control protein 6 homolog (CDC6) (Figure 6a). Moreover, U2OS cells either stably transfected with a mutant form of pRb (K810R) or with *SETD7* knockdown abrogated the cell cycle arrest caused by chemotherapeutics [72]. Supporting this finding, a previous study in mouse embryonic fibroblasts showed that SETD7 methylates PPP1R12A (MYPT1) at K442, a known regulator of pRb, consequently PPP1R12A protein becomes more stable to coordinate the dephosphorylation of pRb [73]. Together, the results suggest that in osteosarcoma cells under DNA damage, SETD7 is needed for cell cycle arrest through methylation of pRb or E2F-1 or PPP1R12A (Figure 6a).

The role of SETD7 as a tumour promoter was reported by four studies. Liu et al. showed that while *SETD7* overexpression in U2OS cells in normal conditions had no effect on HIF-1α, it increased HIF-1α protein levels in hypoxic conditions. Under hypoxic conditions, SETD7 interacts with HIF-1α preventing HIF-1α degradation by the proteasome since *SETD7* knockdown decreased HIF-1α protein but not mRNA, and the proteasome inhibitor MG132 rescued HIF-1α protein expression. *SETD7* knockdown also decreased the expression of several HIF-1α-target genes involved in the glycolysis pathway, including *HK2*, *LDHA* and *PDPK1*, whereas its overexpression caused opposite results. The differential regulation of HIF-1α-target genes by SETD7 was shown to be through enrichment of the H3K4me mark in hypoxia response elements. Stable *SETD7* knockdown with shRNAs significantly decreased cell viability compared to control shRNA after 1 or 2 days in hypoxia. These results suggest that in hypoxia, SETD7 could function as a tumour promoter in osteosarcoma cells by actively regulating HIF-1α-mediated metabolic adaptation and survival [74] (Figure 6b). Lezina et al. showed that in U2OS cells, the inhibition of *SETD7* by shRNA results in G1/S arrest upon DNA damage. SETD7 knockdown in U2OS cells also resulted in an upregulation of E2F-1 protein levels. Moreover, in response to DNA damage, SETD7 differentially modulated the histone modifications (H3K4me1, H3K9me3, AcH3 and AcH4) in the promoters of two E2F-1-dependent genes (*CCNE1* and *TP73*) resulting in activation of *CCNE1* transcription and repression of *TP73* [65] (Figure 6b). Later, Lezina et al. further investigated SETD7 role in DNA damage response (DDR) and DNA repair. Stable knockdown of *SETD7* with shRNA increased the sensitivity of U2OS cells to genotoxic stress by impairing DDR and DNA repair (γ-H2Ax staining). A microarray analysis of wild-type and *SETD7* knockdown cells before and after induction of DNA damage, revealed several genes regulated by SETD7, including *MDM2* whose expression was inversely correlated with *SETD7*. The E3 ubiquitin-protein ligase Mdm2 is known to target p53 for degradation and also for positively regulating E2F-1 activity [41] (Figure 6b). Kontaki et al. showed that in SaOS-2 cells, *SETD7* knockdown increased doxorubicin-induced cell death, whereas *SETD7* overexpression decreased the percentage of cell death upon DNA damage induction. Furthermore, E2F-1 protein levels were highly increased in doxorubicin-treated SaOS-2 cells and in *SETD7* knockdown cells but unchanged in cells overexpressing *SETD7*, suggesting that SETD7 prevents cell death induced by DNA damage by reducing E2F-1 accumulation [38].

In summary, all studies considered the effects of SETD7 in osteosarcoma cell cycle under stress conditions and while some agree that SETD7 prevents cell cycle arrest, others showed the opposite. Most of the studies looked at E2F-1 or expression of its target genes showing that SETD7 can control the levels of E2F-1, its methylation status, regulation and may be required for E2F-1 transcriptional activity, although more studies are needed to understand this mechanism.

#### 3.1.12. Prostate Cancer

Song et al. showed that the nuclear receptor ROR-α2 functions as an oncogene in PCa cell lines PC-3 (androgen receptor (AR)-) and LNCaP (AR+) and that it is activated through methylation at K87 by SETD7. Overexpression of *SETD7* increased binding of ROR-α2 to Tip60, which was weaker in ROR-α2 harbouring the K87R mutation. ROR-α2 K87R significantly decreased *CTNND1* mRNA as compared to ROR-α2 WT and resulted in failure to increase PC-3 cell proliferation and viability. These suggests that SETD7 methylation of RORα2 promotes the recruitment of coactivator complex pontin/Tip60 to ROR-α2 target genes to support ROR-α2 oncogenic potential [75] (Figure 7). Another group published three papers in which they performed SETD7 knockdown in PCa cells [76,77,78] but only analysed the functional consequences in one of them [77]. In the first study, they reported that treating LNCaP cells with the dietary carotenoid astaxanthin increased the expression of the antioxidant *NQO1* mRNA in control cells but not in cells with stable *SETD7* downregulation by shRNAs [78]. In the second study, stable *SETD7* knockdown of PCa cells by shRNA decreased soft agar colony formation ability and increased reactive oxygen species (ROS) and DNA damage. Furthermore, *SETD7* knockdown decreased H3K4me1 enrichment in the *NFE2L2* and *GSTT2* promoter regions and, consequently, their expression in response to challenge with H_2_O_2_. In contrast, treatment with the phytochemicals, PEITC and UA, which upregulated *SETD7* expression, increased the H3K4me1 enrichment at the *NFE2L2* and *GSTT2* promoters in LNCaP cells. Together, these results indicate that SETD7 protects DNA from oxidative damage in LNCaP and PC-3 cell lines possibly by inducing Nrf-2-regulated genes [77] (Figure 7). Later, Wang et al. carried out a transcriptomic analysis of LNCaP cell line with *SETD7* knockdown, with or without PEITC, and found a strong upregulation of the retinoic acid receptor responder 3 (RARRES3) and the antioxidant gene, *SOD2*, when *SETD7* expression was inhibited [76], reinforcing the SETD7 tumour-promoter role through the protection of oxidative stress in PCa cells. Gaughan et al. showed that SETD7 interacts with AR and methylates it at K632. In turn, this enhances *AR* transcriptional activity without affecting its stability or cytoplasmic nuclear shuttling. SETD7 stimulates *PSA* transcription by recruiting AR and enriching H3K4me in its promoter (Figure 7). *SETD7* knockdown with siRNAs decreased LNCaP cell proliferation and increased apoptosis (caspase-3 expression). Further analysis revealed that doxorubicin-induced apoptosis was significantly enhanced by *SETD7* knockdown. Analysis of SETD7 by IHC in 76 cases of human PCa and 24 benign prostate controls showed that nuclear SETD7 was significantly upregulated in epithelial cells and concurrently downregulated in stromal cells from PCa [79]. In agreement with this finding, Gu et al. found an upregulation of *SETD7* in PCa when exploring the TCGA dataset [32]. Interestingly, a whole transcriptomic sequencing of eight castration-resistant prostate cancer samples revealed a frameshift mutation of *SETD7* in one of them [80].

Together, these results showed that in LNCaP and PC-3 cells, SETD7 activity is needed to support cell number increase, either through direct methylation of certain targets (ROR-α2 and/or AR) or H3K4me1 regulation in Nrf-2 gene (*NFE2L2*), since its knockdown decreases cell proliferation and increases apoptosis.

#### 3.1.13. Small Intestine Carcinoma

The in vivo biological function of SETD7 in intestinal tumourigenesis and regeneration was studied using Apc^Min/+/^Setd7^+/−^, Apc^Min/+/^Setd7^−/−^ mice, Setd7^DIEC^ (intestinal epithelial cell-specific *SETD7*-deficient) and Setd7^f/f^ (exon 2 flanked with loxP sites) mice. Apc^Min/+/^Setd7^−/−^ mice had significantly increased lifespans and reduced tumour numbers in the small intestine and colons at endpoint compared with littermate control Apc^Min/+/^Setd7^+/−^ mice. Wnt-target gene (*Axin2*, *Myc* and *Lgr5*) expression was increased in the small intestine tumour tissue compared to normal tissue from both Apc^Min/+/^Setd7^−/−^ and Apc^Min/+/^Setd7^+/−^ mice, but more in tumours from Apc^Min/+/^Setd7^+/−^. This suggests that, during intestinal tumourigenesis, the activation of Wnt/beta-catenin target genes due to Apc loss of function is dependent on SETD7 expression. Consistently, in organoids derived from small intestine tumours in Apc^Min/+^ mice treated with the SETD7 inhibitor (R)-PFI-2 showed reduced initiation of spheres and lower Wnt-dependent gene expression (*Lgr5*, *Axin2* and *Lyz1*). Even though no differences were observed between Setd7^DIEC^ and Setd7^f/f^ mice during dextran sodium sulfate (DSS) or irradiation-induced damage, Setd7^f/f^ mice recovered better than Setd7^DIEC^ mice as measured by body weight, colon length, inflammatory cytokine production, histology score and number of regenerating crypts. Furthermore, *Lgr5* and *Axin2* were downregulated in Setd7^DIEC^, suggesting that SETD7 is not only required for Wnt-dependent intestinal tumourigenesis but also for optimal Wnt/beta-catenin-dependent regeneration following DSS-induced inflammation [8].

#### 3.1.14. Thyroid Carcinoma

Zhao et al. explored the effects of *miR-345-5p*, a negative regulator of SETD7. Zhao et al. found that *miR-345-5p* overexpression inhibited proliferation and promoted apoptosis in the papillary thyroid carcinoma (PTC) TPC1 cell line and the poorly differentiated thyroid gland carcinoma B-CPAP line. The *miR-345-5p* inhibitor produced opposite results. SETD7 overexpression increased cell proliferation and decreased apoptosis while the opposite was observed if SETD7 was knocked down and rescued the *miR-345-5p* anti-tumour effects. The authors also analysed *SETD7* mRNA and protein levels in 15 human PTC and neighbouring normal tissues, with mRNA and protein higher in PTC. A negative correlation between *miR-345-5p* and *SETD7* mRNA expression in PTC tissues was observed [81]. Contrarily, using GEO database (57 PTC and 16 normal), Duan et al. found that *SETD7* expression was downregulated in PTC when compared to normal tissues [42]; however, one should consider the high risk of bias regarding the methodology description of this study.

### 3.2. Regulation of Cancer Biological Processes and Cell Signalling Pathways by SETD7

The cell signalling proteins regulated by SETD7 were predominantly investigated in cell proliferation (Figure 3a), epithelial to mesenchymal transition (EMT)/invasion/metastasis (Figure 3b) and oxidative stress (Figure 3c). Integrating results from the studies described above for the different cancer types, it becomes clear that the most studied pathways targeted by SETD7 were regulated by the retinoblastoma-associated protein (pRb) and transcription factor E2F1 (E2F-1), both under normal conditions and upon treatment with DNA-damaging agents in p53 null or wild-type backgrounds (Figure 5). The effect exerted by SETD7 on target proteins was either to enhance or reduce their degradation by the proteasome, stimulate or inhibit their activity, and/or modulate their subcellular localisation (Table 2). Seventeen studies analysed the effects of SETD7 methylation on protein stability or activity [6,37,38,39,46,48,50,52,53,62,66,71,72,73,75,79] (Table 2), with most of them showing that methylation stabilised or improved target protein function [46,49,50,52,66,71,72,73,75,79]. E2F-1 was the SETD7 target more often studied [31,32,36,38,50,65]. Moreover, signalling pathways, miRNAs and chemical compounds were found to regulate SETD7 expression in different cancers (Table 3). Eleven studies looked into molecules or compounds that regulate SETD7 expression or activity and all, except two [51,67], downregulated SETD7 expression or activity [18,21,34,36,45,54,56,59,81]. Half of these studies identified SETD7 as a tumour promoter and the other half as a tumour suppressor (Table 3). 

#### 3.2.1. Proliferation

Nearly half of the studies showed that knockdown of *SETD7* by siRNAs or the inhibition of its activity with (R)-PFI-2 promoted cell proliferation. In colorectal [6,53] and cervical cancers [48,49], the evidence indicates that SETD7 is a negative regulator of proliferation. However, in PCa [75], BC [45] and small intestine carcinoma [8], SETD7 expression and activity is needed for proliferation (Figure 8a). Higher E2F-1 stability mediated by SETD7 was associated with increases in *CCNE1* [32,65], *CCNA2* and *CDK2* [32] and consequent cell cycle progression and proliferation. However, SETD7 can impair E2F-1 activity indirectly through methylation of pRb [71,72], which in turn is more stable and retains its interaction with E2F-1 to reduce expression of cell cycle progression genes (*DHFR*, *CDK1* and *CDC6*) and stimulate cell cycle arrest [72] and differentiation [71]. Still, it has been shown that Rb/E2F-1 interaction can be inhibited by Mdm2, whose expression is increased by p53 when methylated by SETD7 [41], once again supporting a positive role of SETD7 on cell cycle progression through E2F-1, at least in p53 wild-type tumours (Figure 9). The effects of SETD7 on apoptosis through E2F-1 also appear to be context-specific. SETD7 increases E2F-1 stability, but inhibits E2F-1-dependent transcription of *TP73* in osteosarcoma cells [65]. Alternatively, SETD7 downregulates E2F-1 but also its pro-apoptotic activity [31]. The effects mediated by SETD7 under DNA damaging conditions are unclear. While K185 methylation destabilises E2F-1 to prevent DNA damage-dependent cell death [38], the same methylation stabilises E2F-1 to increase apoptosis through expression of *TP73* and *BIM1* in cells treated with etoposide [50]. The differences between these studies might be due to the usage of cell lines (H1299 and Saos2 vs. U2OS, HCT116 p53^−/−^, and 293T cells) from different tissues and/or DNA-damaging agents with different mechanisms of action.

#### 3.2.2. Differentiation, EMT, Migration, Invasion and Metastasis

The levels of several genes involved in differentiation/stemness [54], EMT [34,36,59,63] and migration and invasion [55,57,64] processes were found to be affected by alterations in *SETD7* expression or activity (Figure 8b). These included cadherin-1 [34,36,59,63,64], vimentin [34,36,59,64], SNAI2 [34,59], cadherin-2 [64] and MMPs [57,64]. All studies showed that, independently of the type of cancer (BlaCa, BC, CRC, GC, HCC and LCa), knockdown of *SETD7* by siRNAs or the inhibition of its activity by (R)-PFI-2 promoted EMT [34,36,59,63] and consequently invasion, migration [34,36,55,57,63,64] and eventually metastasis [36,59]. Therefore, SETD7 appears to suppress these processes.

#### 3.2.3. Hypoxia/Oxidative Stress

Two studies performed in cell lines from different cancer types (BC and CRC) showed that knockdown of SETD7 decreases Nrf-2, consequently decreasing its target genes and increasing ROS accumulation and apoptosis [44,77]. In hypoxia, where ROS levels are increased, SETD7 positively regulates HIF-1α stability and enhances the expression of HIF-1α targets (*HK2*, *LDHA* and *PDPK1*) through enrichment of their promoters with H3K4me1. This results in metabolic remodelling and protection from apoptosis. Thus, in normal conditions, SETD7 targets HIF-1α for degradation operating as a tumour suppressor, but in oxidative stress, SETD7 exerts tumour-promoting effects by maintaining redox homeostasis to protect cancer cells from apoptosis (Figure 8c).

## 4. Conclusions

SETD7 involvement in cancer is evident and its function is relevant for several cancer-related processes, but no consensus regarding its theragnostic potential in cancer can be inferred. First, this review clearly shows that a better description of bioinformatic analysis for databases should be provided. This should include the normalisation methods and clear descriptions of patients’ data along with the cancer histological or molecular subtypes so the information obtained from association studies can be more powerful. The associations found with survival, which pool different cancer subtypes, could simply derive from the association found with grade, stage, etc. Therefore, multivariate analysis should be used to confirm these associations. Second, since the vast majority of SETD7 function involves non-histone protein methylation, the quantitative profiling of SETD7 targets using mass spectrometry and enrichment for lysine methylation in different cancer models (in vivo and in vitro) and in various cellular contexts (hypoxia, genotoxic stress, etc.), is a pressing need in the field. Third, stabilising mutations that prevent methylation can override the SETD7 effect and in some cases, SETD7 activity may be compensated by other methyltransferases, so when analysing the association of SETD7 with the expression of its target proteins, analysis of mutations in the target should be included. Fourth, in many targets, there is a competition between methylation by SETD7 and phosphorylation, so a clear discrimination and control of the signalling context will allow a better understanding of functional studies. While there are data supporting a competition between phosphorylation and lysine methylation on SETD7 substrates, it is likely that competition for other posttranscriptional modifications on the same lysine (i.e., sumoylation, ubiquitination or acetylation) are more prevalent and thus relevant in regulation of cellular signalling by SETD7 and needs further studies. Fifth, the effects observed by SETD7 knockdown may be overridden by other methyltransferases if a longer time window is evaluated in future studies.

With regards to the most common SETD7 functions emerging from this analysis, two-thirds of the literature showed that inhibiting SETD7 function improved response to DNA-damaging agents and given the potential of these finding, efforts to fully establish if modulation SETD7 activity could function as co-adjuvant in patients receiving DNA-damaging therapy are needed. SETD7 controls the pRb/E2F1 axis at many levels and thus could be targeted under specific circumstances where these proteins are mutated, with is linked to therapy resistance. In, line with this, SETD7 methylation of estrogen and androgen receptor enhances their transcriptional activity, so future studies to identify its function in breast and prostate cancer could provide novel insights to treat endocrine-resistant cases. Finally, SETD7 has a consensus role in promoting the differentiation process of several healthy tissues and this role is maintained consistently in cancer cells by inhibiting EMT.

## Figures and Tables

**Figure 1 cancers-14-01414-f001:**
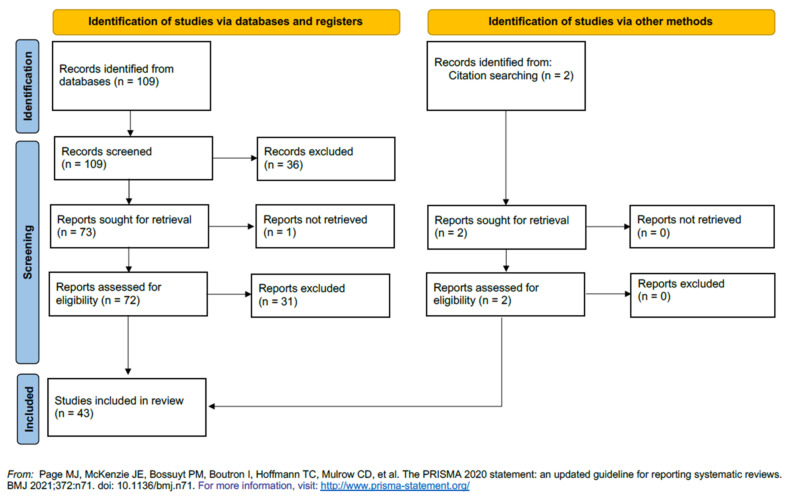
PRISMA flowchart. Flow diagram of the systematic review summarising the literature selection process [24].

**Figure 2 cancers-14-01414-f002:**
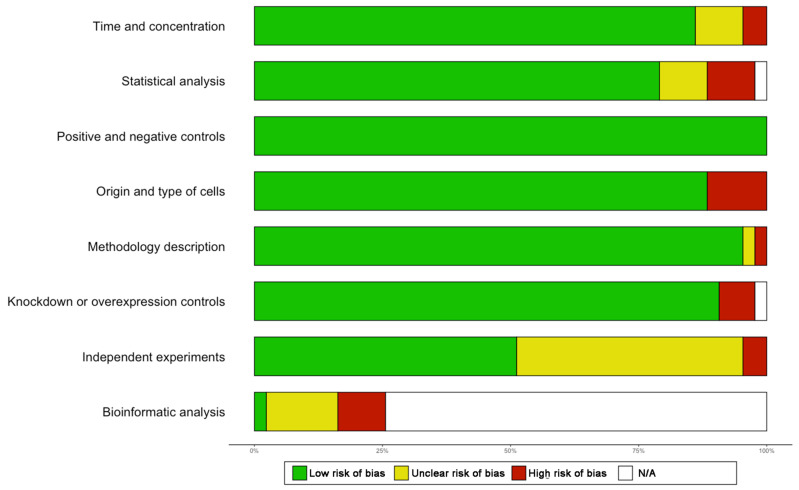
Risk of bias analysis. Summary assessment of the studies included are presented as percentages across all included studies. The following criteria were considered for functional studies: information on time and concentration (chemicals, knockdown, etc.) description and complete report of statistical analysis, usage of positive and negative controls, description of the origin and type of cells used, methodology detail, assessment of gene/protein expression after loss and/or gain of function experiments, usage of independent experiments and description of bioinformatics methodology used for analysis of public datasets or in-house. Green: low risk of bias; yellow: risk of bias is unclear in at least one domain, but no domains with high risk; red: high risk of bias; white: not applicable to the study in question.

**Figure 4 cancers-14-01414-f004:**
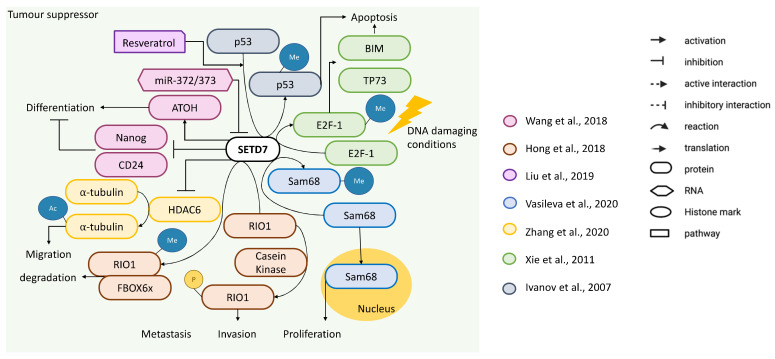
Tumour-suppressor effects of SETD7 in the regulation of differentiation, proliferation, invasion and metastasis in colorectal cancer. miR-372/373 represses SETD7 to reduce differentiation and enhance stemness, since SETD7 downregulates Nanog and CD24 and upregulates ATOH1 [54]. SETD7 inhibits HDAC6 acetylation of alpha-tubulin to prevent migration [55]. SETD7 can inhibit metastasis, invasion and proliferation by methylation of RIO1, which is marked for degradation by interaction with FBXO6; consequently, RIO1 is not phosphorylated by casein kinase [53] to promote invasion. SETD7 methylates Sam68 to prevent its nuclear translocation to induce cell proliferation [6]. Under DNA damaging conditions, SETD7 methylates E2F-1 stabilising it and increasing the expression of the pro-apoptotic genes, *TP73* and *BIM* [50]. Resveratrol [51] induces p53 methylation by SETD7 to increase p53 stability and pro-apoptotic activity [52].

**Figure 6 cancers-14-01414-f006:**
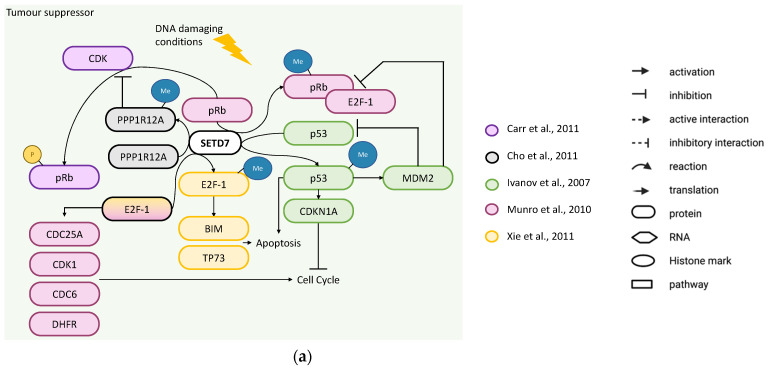
Effect of SETD7 on protein networks in basal and genotoxic stress conditions in osteosarcoma (**a**) Tumour suppressor—Under DNA damaging conditions in osteosarcoma, SETD7 can methylate pRb, increasing its stability and its interaction with E2F-1 to prevent the expression of the E2F-1-target genes *CDK1*, *CDC6*, *CDC25A* and *DHFR* and cell cycle progression [71]. Although this could be prevented by pRb phosphorylation by CDK1 [72]. PPP1R12A can be methylated by SETD7 to coordinate the dephosphorylation of pRb [73]. SETD7 can methylate p53 promoting the expression of *CDKN1A* to inhibit cell cycle progression and *MDM2* [52], which is itself a negative regulator of p53 and pRb/E2F-1 complex [41]. SETD7 can also methylate E2F-1 stabilising it to increase the expression of the pro-apoptotic genes, *TP73* and *BIM* [50]. (**b**) Tumour promoter—Under DNA damaging conditions in osteosarcoma cells, SETD7 downregulates E2F-1 and consequently TP73 and apoptosis [38]. Moreover, in response to DNA damage, SETD7 differentially modulated E2F-1-dependent genes (*CCNE1* and *TP73*) resulting in activation of *CCNE1* transcription [65]. SETD7 reduces the sensitivity of osteosarcoma cells to genotoxic stress by promoting DNA damage repair (DDR) through downregulation of MDM2, which is known to target p53 for degradation and also for positively regulating E2F-1 activity [41]. Under hypoxic conditions, SETD7 interacts with HIF-1α to increase its stability and enhance the expression of HIF-1α targets (HK2, LDHA and PDPK1) through enrichment of their promoters with H3K4me1, to induce HIF-1α-mediated metabolic adaptation and survival [74].

**Figure 7 cancers-14-01414-f007:**
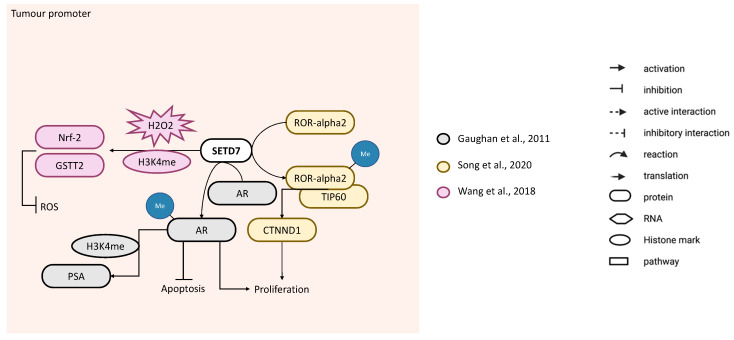
Cellular processes regulated by SETD7 in prostate cancer where it functions as a tumour promoter. SETD7 methylates ROR-alpha2 promoting its binding with TIP60 to induce CTNND1 expression and proliferation [75]. SETD7 can also promote proliferation by methylating to enhance its transcriptional activity. SETD7 stimulates *PSA* transcription by recruiting AR and enriching H3K4me to its promoter [79]. SETD7 increases H3K4me1 enrichment in the *NFE2L2* and *GSTT2* promoter regions and, consequently, their expression in response to challenge with H_2_O_2_ protecting prostate cancer cells from oxidative damage [77].

**Figure 8 cancers-14-01414-f008:**
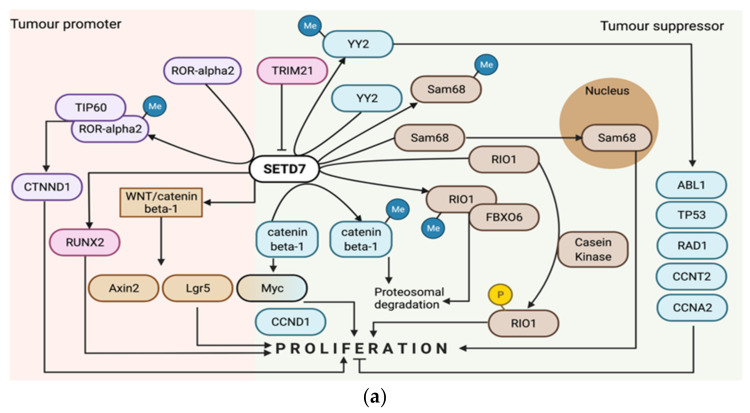
Cellular processes regulated by SETD7. (**a**) Cell proliferation—Positive regulation can be achieved through SETD7 methylation of ROR-alpha2 promoting its binding with TIP60 to induce CTNND1 expression [75]; direct regulation of RUNX2, which can be prevented by TRIM21 [45]; and activation of the WNT/catenin beta-1 pathway to induce AXIN2, LGR5 and MYC expression [8]. Negative regulation can be achieved by SETD7 methylation of catenin beta-1 promoting its proteasomal degradation, thus preventing MYC expression [48]; methylation of RIO1, which is recognised by FBXO6 followed by proteasomal degradation and consequent inhibition of RIO1 phosphorylation by casein kinase to induce cell proliferation [53]; methylation of Sam68 to prevent its nuclear translocation [6] and methylation of YY in YY2 to induce the expression of ABL1, TP53, RAD1, CCNT2 and CCNA2 [49]. (**b**) Differentiation/EMT/Migration/Invasion—SETD7 upregulate cadherin-1 and inhibit vimentin, SNAI2 and cadherin-2 expression preventing EMT, but its mRNA can be methylated by METTL3 at N6 position, which is then recognised by DF2 for degradation [34]. SETD7 inhibits SNAI2 and vimentin and promotes cadherin-1 expression to decrease cell migration and invasion but its mRNA can also be destabilised by SNHG6/hnRNP L [59]. Cadherin-13 positively regulates SETD7 [35]. SETD7 methylates E2F-1 and Dnmt1 inducing their proteasomal degradation to decrease snail expression and inhibition of cadherin-1 and SETD7 expression by reducing snail/Dnmt1 (feedback loop) [36]. SETD7 directly affects cadherin-1 expression [63] through H3K4me enrichment at its promoter [36]. SETD7 inhibits MMP1, MMP7 and MMP9 [57] preventing invasion, and MMP2, TWIST1 and VEGFA and/or JAK/STA pathway [64] and/or alpha-tubulin acetylation by HDAC6 [55] to prevent migration. miR-372/373 represses SETD7 to reduce differentiation and enhance stemness, since SETD7 downregulates Nanog and CD24 and up-regulates ATOH1 [54]. (**c**) *Hypoxia/Oxidative stress—*SETD7 inhibits KEAP1 and induces GSTT2 and Nrf-2 expression through H3K4me enrichment in their promoters supporting the expression of Nrf-2-target genes (TXRND1, ME1, GCLC, GLCM and NQO1) to prevent ROS [77]. ROS can inhibit HIF-1-alpha degradation [82] which is induced by SETD7-mediated methylation [83]. HIF-1-alpha can be demethylated by LSD1 to induce VEGFA and promote tumour growth, angiogenesis and vascularization [61]. On the other hand, under hypoxic conditions, SETD7 interacts with HIF-1-alpha to increase its stability enhance the expression of HIF-1-alpha targets (HK2, LDHA, and PDPK1) through enrichment of their promoters with H3K4me1, to induce HIF-1-alpha-mediated metabolic adaptation and survival [74]. BC—breast cancer; CRC—colorectal cancer; HCC—hepatocellular cancer; LCa—lung cancer; MEFs—mouse embryonic fibroblasts.

**Figure 9 cancers-14-01414-f009:**
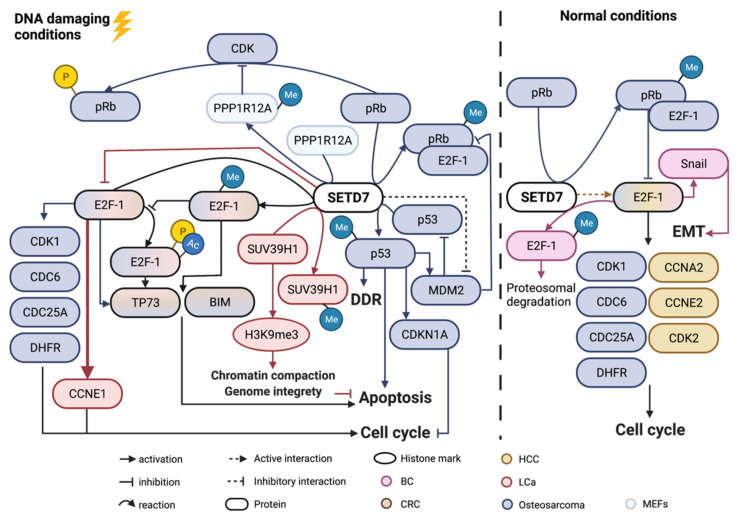
SETD7 influence on the pRb/E2F-1 axis under normal vs. DNA damaging conditions. Under DNA damaging conditions: SETD7 can methylate pRb, increasing its stability and its interaction with E2F-1 [72] to prevent the expression of the E2F-1-target genes *CDK1*, *CDC6*, *CDC25A* and *DHFR* and cell cycle progression [72]; although, this could be prevented by pRb phosphorylation of CDK1 [72]. PPP1R12A can be methylated by SETD7 to coordinate the dephosphorylation of pRb [73]. SETD7 can methylate p53 promoting the expression of *CDKN1A* to inhibit cell cycle progression and *MDM2* [52], which is itself a negative regulator of p53 and pRb/E2F-1 complex [41]. Some studies reported that SETD7 directly methylates E2F-1 preventing E2F-1 stabilisation by acetylation to promote apoptosis mediated by TP73 [38], while others state that E2F-1 methylation by SETD7 stabilises it in increasing the expression of the pro-apoptotic genes, *TP73* and *BIM* [50]. SETD7 can also downregulate E2F-1 activity preventing TP73-mediated apoptosis [31,65], though SETD7 co-activates E2F-1-dependent transcription of *CCNE1* [65]. SETD7 methylates SUV39H1 inhibiting its methyltransferase activity, which results in genome instability and consequent apoptosis [62]. In normal conditions, SETD7 interacts with E2F-1 increasing E2F-1-target gene expression (*CCNA2*, *CCNE2* and *CDK2*) to promote cell cycle [32]. SETD7 can also methylate E2F-1 inducing its proteasomal degradation, which prevents the E2F-1-mediated expression of snail and EMT [36]. SETD7 methylates pRb, increasing its stability and its interaction with E2F-1, thus preventing the expression of the E2F-1-target genes (*CDK1*, *CDC6*, *CDC25A* and *DHFR)* and cell cycle progression [71]. The colour of the lines refers to the type of cancer where the reactions occur. BC—breast cancer; CRC—colorectal cancer; HCC—hepatocellular carcinoma; LCa—lung cancer; MEFs—mouse embryonic fibroblasts.

**Table 1 cancers-14-01414-t001:** Research question formulated as PICO framework.

Population	Intervention	Comparison	Outcome
Human	SETD7 overexpression SETD7 knockdown SETD7 knockout SETD7 mutation Chemical inhibition of SETD7 methyltransferase activity	Control transfected wild type	Any
Mouse		Vehicle/solvent	

**Table 2 cancers-14-01414-t002:** Effects of SETD7-mediated methylation in target proteins.

Study	Cancer	SETD7-Target	Methylation Site	Effect	Target Protein Function	SETD7 Function
Kontaki et al., 2010 [38]	LCa	E2F-1	K185	Protein destabilisation	Tumour suppressor	Tumour promoter
Fu et al., 2016 [66]	LCa	GLI3	K436 and K595	K436—protein stabilisation; K595—improves protein binding to DNA	Tumour promoter	
Gaughan et al., 2011 [79]	PCa	AR	K632	Enhances AR activity	Tumour promoter	
Song et al., 2020 [75]	PCa	ROR-a2	K87	Increases ROR-a2 activity	Tumour promoter	
Shen et al., 2015 [48]	Cervical cancer	Catenin beta-1	K180	Protein destabilisation	Tumour promoter	Tumour suppressor
Wu et al., 2017 [49]	Cervical cancer	YY2	K247	Protein stabilisation	Tumour suppressor	
Vasileva et al., 2020 [6]	CRC	Sam68	K208	Increases cytoplasm localisation (impairing Sam68 nuclear activity)	Tumour promoter (in nucleus)	
Hong et al., 2018 [53]	CRC and GC	RIO1	K411	Protein destabilisation	Tumour promoter	
Xie et al., 2011 [50]	CRC and osteosarcoma	E2F-1	K185	Protein stabilisation	Tumour suppressor	
Wang et al., 2013 [62]	LCa	SUV39H1	K105 and K123	Inhibited SUV39H1 activity	Tumour promoter	
Kim et al., 2016 [83]	LCa	Hif-1-a	K32	Protein destabilisation	Tumour promoter	
Ivanov et al., 2007 [52]	Osteosarcoma	p53	K372	Protein stabilisation	Tumour suppressor	
Munro et al., 2010 [71]	Osteosarcoma	pRb	K873	Protein stabilisation	Tumour suppressor	
Carr et al., 2011 [72]	Osteosarcoma	pRb	K810	Protein stabilisation	Tumour suppressor	
Estève et al., 2009 [37]	Colorectal cancer	DNMT1	K142	Protein destabilisation	NA	NA
Subramanian et al., 2008 [46]	BC	ERa	K302	Protein stabilisation	NA	NA
Cho et al., 2011 [73]	MEFs	PPP1R12A	K442	Protein stabilisation	NA	NA

BC—breast cancer; CRC—colorectal cancer; GC—gastric cancer; LCa—lung cancer; MEFs—mouse embryonic fibroblasts; PCa—prostate cancer.

**Table 3 cancers-14-01414-t003:** Regulation of SETD7.

Study	Cancer	SETD7-Regulator	Effect	SETD7 Function
Montenegro et al., 2016 [36]	BC	E2F-1 through Dnmt1and SNAI1	Downregulate	Tumour suppressor
Xie et al., 2020 [34]	BlaCa	METTL3/YTHDF2 m^6^A axis	Downregulate	
Wang et al., 2018 [54]	CRC	miR-372/373	Downregulate	
Liu et al., 2019 [51]	CRC	Resveratrol	Upregulate	
Wang et al., 2021 [59]	HCC	SNHG6	Destabilises	
Francis et al., 2012 [18]	CRC	610930-N	Inhibits activity	Tumour promoter
Takemoto., 2016 [21]	BC	Cyproheptadine	Inhibits activity	
Si et al., 2020 [45]	BC	TRIM21	Degradation	
Zhou et al., 2015 [56]	EOC	miR-153	Downregulate	
Zhao et al., 2020 [81]	PTC	miR-345-5p	Downregulate	
Meng F et al., 2020 [67]	LCa	LINC01194	Upregulate	

BC—breast cancer; CRC—colorectal cancer; GC—gastric cancer; LCa—lung cancer; MEFs—mouse embryoinic fibroblasta; PCa—prostate cancer.

## Data Availability

Not applicable.

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
