# Peer review of "A Systematic Review to Define the Multi-Faceted Role of Lysine Methyltransferase SETD7 in Cancer"

_cancers, 2022, doi:10.3390/cancers14061414_

Round 1
Reviewer 1 Report
The authors have addressed all my concerns, and the manuscript is acceptable.
Author Response
We appreciate the time dedicated to review our work.
Reviewer 2 Report
The authors have addressed the majority of the points raised. The interplay between phosphorylation and lysine methylation on SETD7 substrates is still given great prominence, when it seems much more likely that interplay between/competetition between modifications on the same lysine would be much more likely and prevalent e.g. methylation vs sumoylation/ubiquitination/acetylation. This could easily be added as a sentence in the discussion.
Author Response
We appreciate the time dedicated to review our work. We have addressed the constructive critic by adding the following sentence to the conclusion: "While there is data supporting a competition between phosphorylation and lysine methylation on SETD7 substrates, it is likely that competition for other postranscriptional modifications on the same lysine (i.e symoylation, ubiquitination or acetylation) are more prevalent and thus relevant in regulation of cellular signalling by SETD7 and needs further studies." (please see lines 1012-1016).